# ReporterSeq reveals genome-wide dynamic modulators of the heat shock response across diverse stressors

Brian D Alford[1†], Eduardo Tassoni-Tsuchida[1,2†], Danish Khan[1], Jeremy J Work[1], Gregory Valiant[3], Onn Brandman[1]*

[1]Department of Biochemistry, Stanford University, Stanford, United States; [2]Department of Biology, Stanford University, Stanford, United States; [3]Department of Computer Science, Stanford University, Stanford, United States

**Abstract** Understanding cellular stress response pathways is challenging because of the complexity of regulatory mechanisms and response dynamics, which can vary with both time and the type of stress. We developed a reverse genetic method called ReporterSeq to comprehensively identify genes regulating a stress-induced transcription factor under multiple conditions in a time-resolved manner. ReporterSeq links RNA-encoded barcode levels to pathway-specific output under genetic perturbations, allowing pooled pathway activity measurements via DNA sequencing alone and without cell enrichment or single-cell isolation. We used ReporterSeq to identify regulators of the heat shock response (HSR), a conserved, poorly understood transcriptional program that protects cells from proteotoxicity and is misregulated in disease. Genome-wide HSR regulation in budding yeast was assessed across 15 stress conditions, uncovering novel stress-specific, time-specific, and constitutive regulators. ReporterSeq can assess the genetic regulators of any transcriptional pathway with the scale of pooled genetic screens and the precision of pathway-specific readouts.

*For correspondence:
onn@stanford.edu

†These authors contributed equally to this work

Competing interests: The authors declare that no competing interests exist.

## Introduction

The heat shock response (HSR) is a conserved stress response that shields cells from cytoplasmic proteotoxicity by increasing the expression of protective proteins (*Lindquist, 1986*; *Morimoto, 2011*). The HSR is driven by the transcription factor Hsf1, a trimeric protein that promotes expression of chaperones and other cellular components that refold misfolded proteins or target them for degradation (*Akerfelt et al., 2010*). A diverse array of stressors activate the HSR, including heat, ethanol, oxidative stressors, and amino acid analogs (*Morano et al., 2012*). In these conditions, HSR activity can follow diverse trajectories, with both transient or prolonged activity observed (*Sorger, 1990*). Furthermore, aberrant HSR activation has been linked to neurodegeneration (too little HSR activation; *Campanella et al., 2018*) and cancer (too much HSR activation; *Whitesell and Lindquist, 2009*).

The complexity of the HSR has led to a multitude of models for how the HSR is regulated (*Anckar and Sistonen, 2011*). These include negative regulation in which the chaperones Hsp70 (*Krakowiak et al., 2018*; *Zheng et al., 2016*), Hsp90 (*Ali et al., 1998*; *Zou et al., 1998*), and Hsp60 (*Neef et al., 2014*) bind and inhibit Hsf1 directly. Signaling pathways have also been demonstrated to modulate Hsf1 activity through phosphorylation by kinases such as Snf1, Yak1, and Rim15 (*Hahn and Thiele, 2004*; *Lee et al., 2013*, *Lee et al., 2008*). Additionally, post-transcriptional models have been proposed, such as HSR-regulated mRNA half life (*Heikkinen et al., 2003*) or regulation of the translation efficiency of heat shock mRNAs (*Zid and O'Shea, 2014*). Yet the extent to

which these and other mechanisms drive the HSR under diverse stressors is poorly understood, limiting our ability to understand and remedy the HSR in disease states (*Neckers and Workman, 2012*).

Reverse genetic studies of the HSR have been useful in discovering both new protein quality control machinery in the cell (*Brandman et al., 2012*), as well as new methods of regulating the HSR (*Raychaudhuri et al., 2014*). However, existing reverse genetic approaches to measure specific pathway activity require measuring the effect of each gene separately or enriching cells according to pathway activity, making it impractical to test multiple environmental conditions or make time-resolved measurements.

To address this methodological deficit and knowledge gap, we developed ReporterSeq, a pooled, genome-wide screening technology that can measure the effect of genetic perturbations with the scale of pooled genetic screens and the precision of pathway-specific readouts (e.g. quantitative PCR [qPCR], fluorescence, luciferase reporters). ReporterSeq measures pathway activity under a specific perturbation through levels of a corresponding barcode. Thus, ReporterSeq does not require cell sorting and instead relies only upon measuring barcode frequencies at a single genetic locus of a pooled sample. This allows dozens of samples to be collected on the same day, prepared for sequencing, and then read out in a single high-throughput sequencing run. Because ReporterSeq uses RNA levels as a direct readout of transcriptional activity, pathway activity can be measured even in conditions in which protein synthesis is compromised.

We performed ReporterSeq in the budding yeast *Saccharomyces cerevisiae* under basal growth conditions and 13 stress conditions to identify general and stress-specific regulators of the HSR. This elucidated the roles of known regulators, like the Hsp70 chaperone system and Snf1 kinase complex, as well as novel regulators, including Gcn3 and Asc1, in responding to diverse stressors. A time course of HSR activation in response to heat stress revealed distinct regulatory mechanisms in the early and late stages of heat stress. We further investigated how loss of Gcn3 regulates the HSR and found that, contrary to its canonical role in stress-dependent inhibition of translation, Gcn3 is required for efficient translation of both Hsf1-dependent and Hsf1-independent stress-dependent reporter genes under arsenite stress. In addition to elucidating HSR regulation, our work demonstrates that ReporterSeq is a generally applicable tool to dissect the genetic basis for the activity of any transcription factor in a quantitative, scalable, and time-resolved manner.

## Results

### A pooled strategy to measure expression of a specific gene under genome-wide, CRISPR perturbations without cell enrichment

ReporterSeq measures how endogenous genes affect the expression of an exogenous reporter gene. It accomplishes this through pairing a genetic perturbation with barcode levels from a pathway-specific readout (*Figure 1*). Though ReporterSeq can be implemented with any encodable genetic perturbation (e.g. RNAi, CRISPR knockouts), we used CRISPRi (*Gilbert et al., 2013*) as a perturbation to lower expression of each gene and paired it with a barcoded mRNA encoding GFP downstream of an Hsf1-responsive synthetic promoter built upon a 'crippled' *CYC1* promoter sequence derived from a 225 nucleotide fragment of the *CYC1* promoter (*Brandman et al., 2012*; *Guarente and Mason, 1983*). This synthetic promoter, previously used in a fluorescence-based HSR screen (*Brandman et al., 2012*; *Guarente and Mason, 1983*), is sensitive to Hsf1 activity as well as proteins that may regulate the *CYC1* sequence fragment (note that no transcription factors other than Hsf1 are known to bind to the sequence). To maximize dCas9-Mxi1 (a transcription inhibitor) and sgRNA expression, we expressed dCas9-Mxi1 and the ReporterSeq cassette on two separate high copy plasmids that were coexpressed in each cell. We verified that the genetic knockdowns could modulate the heat shock reporter by measuring the effects of known HSR regulators (*Brandman et al., 2012*) on reporter output (*Figure 1—figure supplement 1A*). We therefore concluded that CRISPRi knockdowns were effective in our system and capable of altering HSR reporter activity.

The first step of ReporterSeq is to synthesize a diverse library of constructs, each containing one sgRNA and one barcode driven by a promoter of interest. The library is then introduced into cells such that each cell contains one construct. Cells are grown in the desired conditions and then harvested at the desired timepoints. RNA and DNA are harvested from the cells and the counts of each

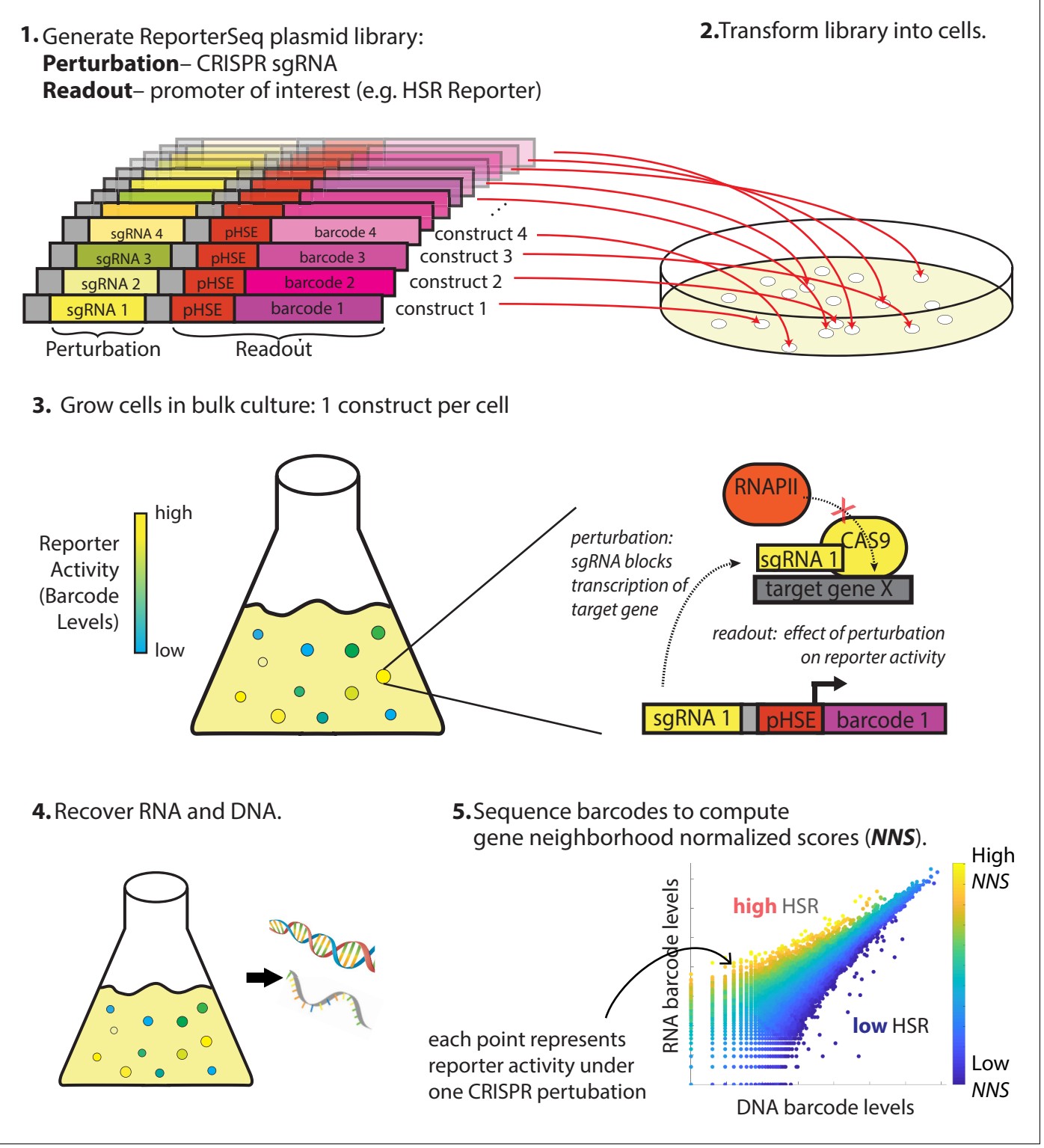

**Figure 1.** Overview of ReporterSeq method. Example given for measuring the heat shock response (HSR) using a heat shock element (HSE)-driven promoter and CRISPRi to perturb gene expression. ReporterSeq measures the effect of genetic perturbations with the throughput of pooled screens and the precision of pathway-specific, transcriptional readouts.

The online version of this article includes the following figure supplement(s) for figure 1:

**Figure supplement 1.** Diversity of ReporterSeq library.

barcode are tallied using deep sequencing. The RNA counts for a given barcode are proportional to the total transcriptional output of all cells containing that barcode, while the DNA barcode counts are proportional to cell numbers. Thus, the RNA/DNA ratio of a given barcode reflects the activity of the reporter under the influence of the specific CRISPR knockdown corresponding to that barcode. This ratio is used to identify the effect of a given knockdown on the HSR in a single condition. In situations where the same pool of yeast is divided and exposed to multiple conditions, RNA counts between these samples can be directly compared to compute gene–stressor interactions (discussed below).

Our library targeted each gene in the genome, including genes producing non-coding RNAs, with up to 12 sgRNAs and paired multiple random barcodes with each sgRNA. Multiple sgRNAs per gene were necessary due to variability in efficacy of each knockdown and to minimize the influence of off-target effects from any one guide RNA. Using multiple barcodes for each sgRNA minimized the effects of barcode bias on stability of the reporter mRNA. We cloned a library of approximately 1 million random barcode-sgRNA pairs and performed paired-end high-throughput sequencing on the library to identify these pairs. Paired-end sequencing revealed that most genes were targeted in the library by all 12 designed sgRNAs (*Figure 1—figure supplement 1B*). Additionally, there was a wide distribution of barcodes associated with each sgRNA, with the mean representation being approximately 14 barcodes per sgRNA (*Figure 1—figure supplement 1C*). Essential genes were only weakly depleted from the library, reflecting that our screen measures the effect of knockdown and that many sgRNAs do not lead to effective knockdown (*Figure 1—figure supplement 1D*). Note that after multiple generations of growth, we expect that only one barcoded construct will be expressed in each cell even in the rare event that two constructs initially enter the same cell. Thus, the plasmid library was poised to measure the effect of each genetic knockdown on the output of the HSR reporter.

## ReporterSeq reveals regulators of the HSR in basal conditions

We first used the ReporterSeq library to measure the effect of each genetic knockdown on the HSR reporter in untreated, log-phase yeast. The full-genome library was transformed into >100,000 cells, resulting in a subset of the plasmid library expression in cells. RNA and DNA barcodes were sequenced and matched to their corresponding sgRNAs, revealing that 91% of genes were represented by six or more sgRNAs where both RNA and DNA counts were above 10 (*Figure 1—figure supplement 1E*) and 74% of these sgRNAs were represented by two or more barcodes (*Figure 1—figure supplement 1F*). To test whether the RNA/DNA ratio of barcodes reflected activation of the HSR under a specific perturbation, we assessed the effect of two predicted regulators of our synthetic HSR reporter (*Figure 2A*). sgRNAs targeting *SSA2*, an Hsp70 chaperone that binds Hsf1 and inhibits the HSR (*Krakowiak et al., 2018*; *Zheng et al., 2016*), resulted in a higher than average RNA/DNA ratio as expected. Conversely, sgRNAs targeting *CYC1*, the gene from which our synthetic reporter was derived and shares significant homology with, lowered the RNA/DNA ratio compared to average. Thus, the DNA/RNA ratio for a single barcode reflects the HSR reporter's activity under a specific genomic perturbation. Note that since we do not explicitly measure knockdown efficiency of the CRISPR constructs, our results cannot rule out the effect of any gene on the HSR.

To facilitate comparisons between many data points, we developed a scoring method called the neighborhood normalized score (*NNS*) that combined the information from the multiple barcodes and corresponding sgRNAs which target a single gene into a single score. Briefly, each barcode was assigned an *NNS*, representing how extreme the RNA/DNA ratio is for the barcode, relative to its 'neighborhood' of barcodes with similar levels of expression. The magnitude of the score for a given barcode corresponds to the number of standard deviations from the mean, where both the standard deviation and mean are computed based on this neighborhood. The sign of the score indicates the direction of the effect on the HSR (positive *NNS* indicate that the knockdown increases the HSR, while negative *NNS* indicate the knockdown decreases the HSR). Most barcodes have *NNS* near 0, reflecting the fact that most perturbations do not significantly affect the HSR. The *NNS* for each gene was tallied as the average barcode *NNS* over all sgRNAs and barcodes corresponding to the gene in question. (See the Methods and methods section for details on how the *NNS* is calculated.) To give two examples, *SSA2* knockdown, had the fourth highest *NNS* at 7.8, indicating that the knockdown of *SSA2* likely increased reporter activity (*Figure 2B*), with the corresponding barcodes being 7.8 standard deviations above the mean, on average. *CYC1*, conversely, had the third lowest

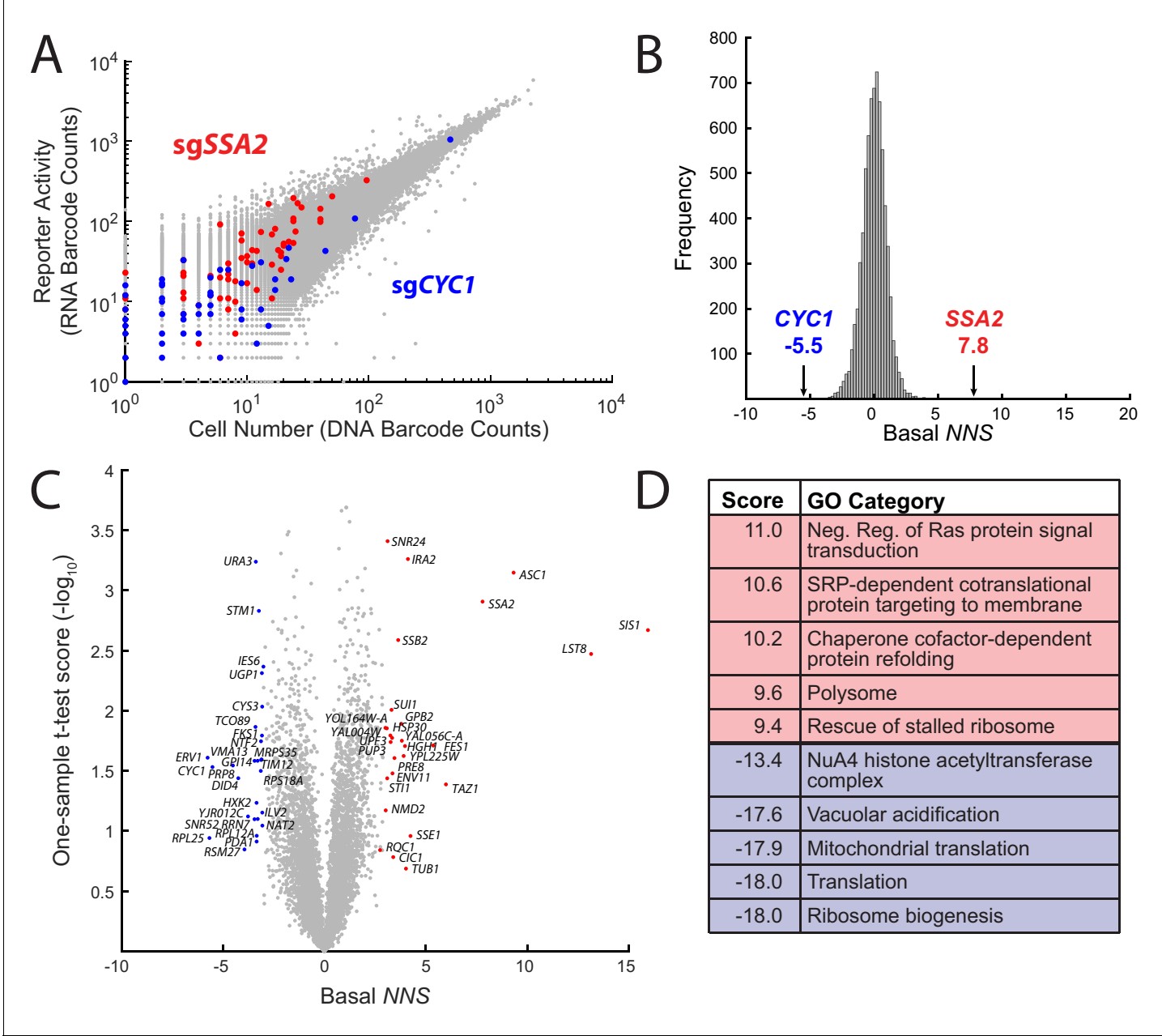

**Figure 2.** ReporterSeq reveals genome-wide basal regulators of the heat shock response. (**A**) Comparison of DNA and RNA barcode counts from wild-type, untreated yeast. Red dots indicate barcodes that correspond to sgRNAs targeting the Hsp70 chaperone, *SSA2*. Blue dots indicate barcodes that correspond to sgRNAs targeting *CYC1*, a gene that shares significant homology with the Hsf1-driven synthetic reporter. (**B**) Histogram of the basal *NNS* for sgRNAs targeting each gene. *NNS* for *SSA2* and *CYC1* gene targets are indicated. (**C**) Plot of the basal *NNS* for every genetic knockdown. Outliers and genes mentioned in the text are labeled. (**D**) Table of the gene ontology (GO) categories with the five highest and five lowest scores in the basal screen.

The online version of this article includes the following figure supplement(s) for figure 2:

**Figure supplement 1.** Barcode and sgRNA specific effects on HSR reporter.

**Figure supplement 2.** Top hits are dependent upon HSE sequence of HSR reporter.

**Figure supplement 3.** Comparisons of ReporterSeq screen with previous HSR screen.

*NNS* of −5.5, meaning that it likely decreased the reporter activity. All gene *NNS* are provided in supplementary data file 1 and fold changes for top and bottom 20 *NNS* hits (e.g. *SIS1* = ~100% increase, *ERV1* = ~30% reduction) are provided in supplementary data file 2. Individual sgRNA *NNS* as well as barcode *NNS* for the top scoring sgRNA are shown for *CYC1*, *SSA2*, and, another top hit, *SIS1* (*Figure 2—figure supplement 1*). Individual sgRNA counts and *NNS* for all genes are provided in supplementary data file 3 (basal *NNS*) and supplementary data file 4 (all stress conditions).

We quantified the effect of each genetic knockdown on the basal HSR reporter activity using our scoring metric and used a one-sample t-test to quantify the reproducibility of each *NNS* in the four replicates that we performed (*Figure 2C*). CRISPRi knockdown of chaperones or cochaperones *SSA2*, *SIS1* (*NNS* = 16.0), *FES1* (*NNS* = 4.1), *SSE1* (*NNS* = 5.3), *HGH1* (*NNS* = 4.0), *SSB2* (*NNS* = 3.6), and *STI1* (*NNS* = 3.1) strongly activated the HSR. Indeed, the Hsp40 *SIS1* was the strongest basal suppressor of the HSR. Additional strong HSR suppressors include the negative regulators of the Ras/cAMP pathway *GPB2* (*NNS* = 3.8) and *IRA2* (*NNS* = 4.1), the mitochondrial lyso-phosphatidylcholine acyltransferase *TAZ1* (*NNS* = 6.0), the nonsense-mediated decay regulators *UPF3* (*NNS* = 3.3) and *NMD2* (*NNS* = 3.0) and the small ribosomal protein *ASC1* (*NNS* = 9.3). *LST8* (*NNS* = 13.1) was also a strong hit in the screen, likely because it shares a promoter with *SIS1* and thus silenced by the same sgRNAs (*Pincus et al., 2018*). Our implementation of ReporterSeq cannot distinguish between adjacent genes that may be targeted by the same sgRNAs, a general shortcoming of CRISPRi. In addition to *CYC1*, several mitochondrial genes (*ERV1* **NNS** = −5.8, *RSM27* **NNS** = −3.9, and *TIM12* **NNS** = −3.1) and translation genes (*RPL25* **NNS** = −5.6, *RPL12A* **NNS** = −3.3, and *RPS18A* **NNS** = −3.1) strongly lowered the HSR when impaired, consistent with previous observations that knockout of mitochondrial genes lowers the HSR (*Brandman et al., 2012*). To test whether top hits in the screen were dependent on the Hsf1 binding sites in our reporter rather than the crippled *CYC1* sequence, we compared the effect of deleting three of the top hits (*SSA2*, *HSC82*, and *ASC1*) on the HSR reporter versus a crippled *CYC1* reporter without the Hsf1-specific binding sequence (*Figure 2—figure supplement 2*). All deletions activated the HSR reporter without activating the crippled *CYC1* reporter, which has very low basal activity, suggesting that most hits in the screen are specific to Hsf1.

In order to identify processes that regulate the HSR, we analyzed which gene ontology (GO) categories (*Ashburner et al., 2000*; *The Gene Ontology Consortium, 2019*) were enriched for genes causing high or low activation of the HSR when knocked down. We assigned each GO category a score based on the Student's t-test between the scores of all genes and those of the specific category (supplementary data file 5). As with the interaction score, the sign indicates the direction of the effect of the knockdown on the reporter activity, and the magnitude of this score indicates the confidence. Many of the GO category scores followed trends largely consistent with previous studies, with chaperone, protein folding, and protein quality control complexes such as the ribosome-associated quality control complex (RQC) amongst the most highly scoring GO categories (*Figure 2D*). The highest scoring GO category was negative regulation of Ras signal transduction, indicating that high Ras activity leads to high HSR activity. This is consistent with a previous study showing an inverse correlation between the HSR and the Ras/PKA-inhibited Msn2/4 pathway (*Brandman et al., 2012*) and studies concluding that the PKA activity induces the HSR (*Murshid et al., 2010*; *Verma et al., 2020*) The lowest scoring categories included mitochondria and translation, consistent with results of a previous HSR screen (*Brandman et al., 2012*). Knockdowns of genes involved in vacuolar acidification, a process important for many cellular activities such as protein sorting and pH maintenance (*Yamashiro et al., 1990*) as well as regulation of the HSR (*Triandafillou et al., 2018*) also caused a significant downregulation of the HSR.

## Interactions between heat stress and genes that regulate the HSR

To determine which genes affect the cell's ability to activate the HSR in response to heat stress, we sought to quantify the interaction between the effects of heat stress and those of each genetic knockdown (a 'stressor–gene' interaction) on the output of the HSR reporter (*Figure 3A*), similar to how genetic interactions ('gene–gene' interactions) are conceptualized (*Collins et al., 2007*; *Tong et al., 2001*). Stressor–gene interactions are based on a comparison of the reporter output of the combined condition relative to that of the knockdown and stress independently. A stressor–gene interaction *NNS* of zero denotes that knockdown of the gene and the stressor itself have independent effects on the HSR. A positive *NNS* denotes that the combination of gene and stress

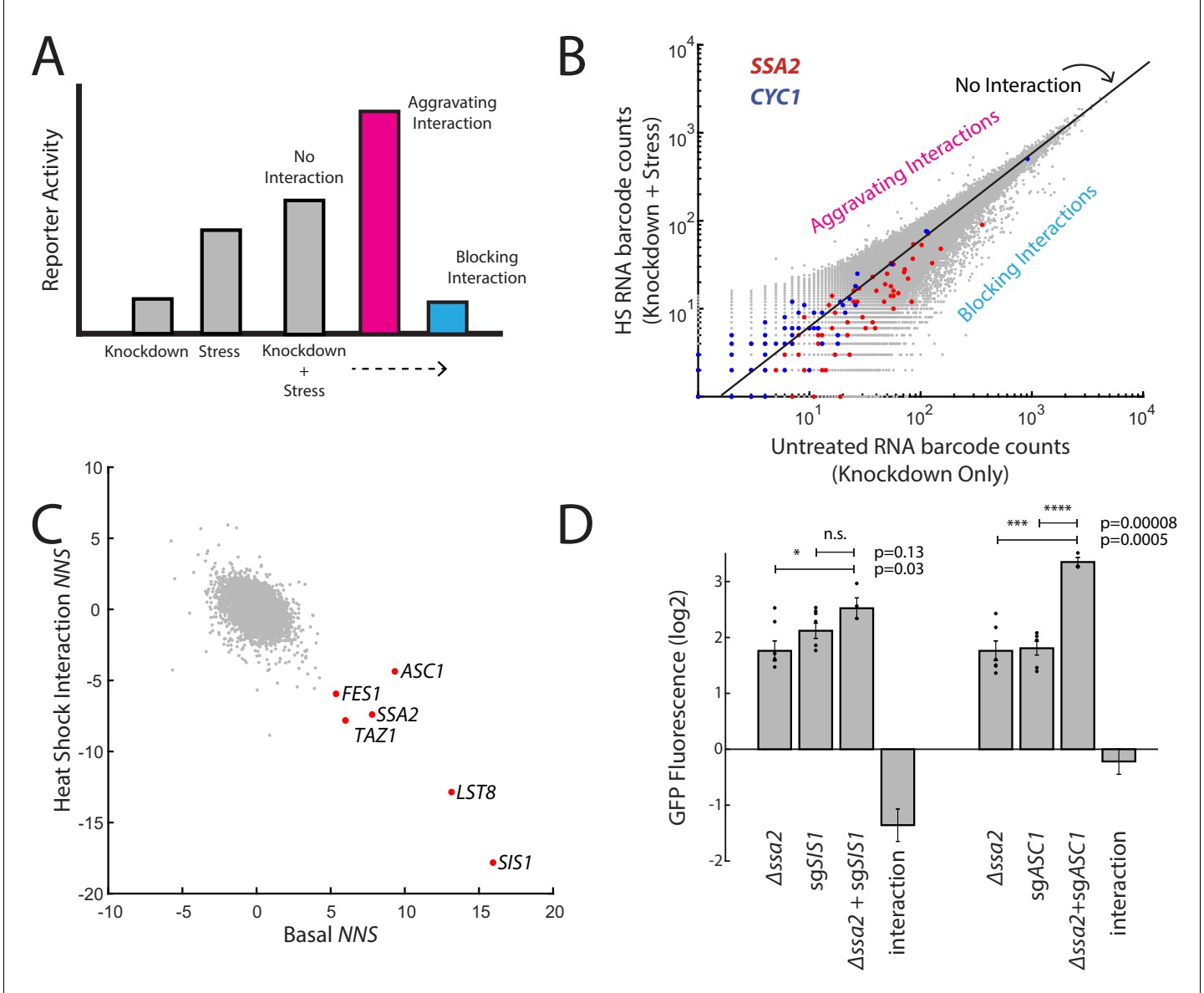

**Figure 3.** ReporterSeq reveals how genes interact with heat stress. (**A**) A schematic of gene–stressor interactions. The combination of a gene perturbation and stressor can be additive (no interaction, suggesting independence), higher than expected (an aggravating interaction), or lower than expected (a blocking interaction). (**B**) RNA barcode counts of untreated yeast compared to those of heat-stressed yeast from the same sample. No interaction, aggravating, and blocking interactions are indicated. Red dots indicate barcodes that correspond to sgRNAs targeting *SSA2*. Blue dots indicate barcodes that correspond to sgRNAs targeting *CYC1*. (**C**) Basal *NNS* for each gene versus the heat shock interaction *NNS* for each gene. Genes with high basal *NNS* are labeled. (**D**) Genetic interactions for Δ*ssa2*-sg*SIS1* and Δ*ssa2*-sg*ASC1* gene pairs. GFP fluorescence of the HSR reporter in the *ssa2* knockout alone, an sgRNA alone, and both perturbations together are displayed, each relative to wild-type cells containing the reporter. The genetic interaction is the observed combined log2 reporter activity minus the reporter activity in each individual genetic perturbation. Error bars are standard errors of three to six replicates. p-values are calculated based on an unpaired t-test: n.s.p>0.05; *p<0.05; **p<0.01; ***p<0.001; ****p<0.0001.

resulted in a superadditive effect on the HSR (an 'aggravating' interaction), while a negative *NNS* denotes that the combination results in a subadditive effect on the HSR (a 'blocking' interaction). A positive or negative interaction *NNS* suggests that a stressor and gene are related in the context of the HSR. Traditional methods to compute interaction scores require three measurements: the phenotype of each individual perturbation and the phenotype of the double mutant. By assuming that (1) the majority of genetic knockdowns do not interact with the stressor and (2) short stressor

treatments that impair cell growth (relative to the ~2.5 hr doubling time of unstressed yeast in the growth conditions used) do not significantly alter cell counts for a specific CRISPRi perturbation, we computed stressor–gene interactions by comparing RNA levels between stressed and unstressed conditions from the same yeast sample using the same scoring metric as with the untreated RNA/DNA ratios. This allowed us to compute the stressor–gene interactions without calculating the absolute induction of the gene (its RNA/DNA ratio), reducing noise arising from DNA normalization and increasing robustness by performing more comparisons than traditional methods for computing gene–gene interactions.

We then examined which genes showed strong interactions with a heat stress of 20 min at 39°C. As an example of the interaction *NNS* in practice, we first examined *SSA2* and *CYC1*. *SSA2* had a blocking interaction with heat (**NNS** = −7.4) whereas *CYC1* had an aggravating interaction with heat stress (**NNS** = 2.1; *Figure 3B*). This suggests (1) that heat stress and *SIS1* depletion activate the HSR through common mechanisms and (2) that targeting both the *CYC1* gene as well as the HSR reporter directly (because the reporter is built upon the *CYC1* promoter sequence) do not reduce its inducibility by heat. Indeed, induction was higher than expected in knockdown of *CYC1* (a positive interaction) and, generally, heat stress interaction *NNS* and the basal knockdown *NNS* were negatively correlated (r = −0.45), suggesting that knockdowns basaly increasing the HSR generally blocked HSR activation in heat and those decreasing basal HSR resulted in a bigger HSR increase during heat stress.

One knockdown target that blocked the response to heat less than would be predicted based on the general trend for strong HSR inducers was *ASC1* (*Figure 3C*). This suggests that activation of the HSR by *ASC1* may be through mechanisms independent of the other genes that strongly activated the HSR, such as *SIS1* and *SSA2*. To test this, we measured the genetic interaction between combined genetic perturbations (deletions combined with CRISPRi) to determine if these genes belonged in a shared or independent pathway of activating the HSR (*Figure 3D*). As expected, *SIS1* (encoding a cochaperone of Hsp70) and *SSA2* (encoding an Hsp70) had a blocking interaction with each other, as the combination of the two perturbations was 65% of the expected GFP levels (quantified as log GFP levels). By contrast, *ASC1* was independent of *SSA2*, with the combination of genetic perturbations having an almost completely additive effect on the reporter (GFP levels 94% of expected levels). This suggests that lowering Asc1 levels may activate the HSR through a separate mechanism than lowering Hsp70 levels.

## Distinct genes regulate the early and late phases of the HSR

The pooled format of ReporterSeq and very short sample collection time allowed us to measure the effect of each knockdown on the HSR in a time-resolved way with a precision that genome-wide approaches generally cannot achieve. We collected cells at 10, 20, 40, and 120 min after transition to 39°C in duplicate and isolated RNA to calculate stressor–gene interactions each time point (*Figure 4A*). CRISPR knockdown of two sample genes, *SIS1* and *SSA2*, were not affected by heat shock duration, suggesting that knockdown efficiency does not change during the time course (*Figure 4—figure supplement 1*). Many genes had similar interaction *NNS* for the first and last timepoints. For example, knocking down *SIS1* or *TPS1* blocked the HSR throughout the time course (*Figure 4B*). Tps1 is required for synthesis of trehalose and has been demonstrated to be required for induction of the HSR under heat shock (*Conlin and Nelson, 2007*). Other interactions, however, greatly depended on the length of heat shock. Knockdown of the Ras gene, *RAS2*, caused an aggravation of the HSR at 10 and 20 min, had no effect at 40 min, and then subsequently blocked the HSR at 120 min. Knockdown of *HSF1* had a strong blocking effect at 10 min that tapered to near zero by 40 min. Conversely, knockdown of *HSP42, BTN2*, or *OPI10* had relatively little effect until 40 and 120 min, at which point the HSR became hyper-induced. These genes have been reported to be critical for survival in heat shocked cells, with Hsp42 and Btn2 sequestering toxic proteins (*Haslbeck et al., 2004*; *Malinovska et al., 2012*; *Specht et al., 2011*) and the human and *Schizosaccharomyces pombe* orthologs of Opi10 ('Hikeshi') facilitating the stress-induced import of Hsp70 into the nucleus (*Kose et al., 2012*; *Song et al., 2015*). The delay in the effects of these factors likely represents the time in which cellular damage accumulates under heat shock. Curiously, the nearly identical, highly expressed HSP70 paralogs *SSA1* and *SSA2* had strikingly different interaction profiles over time, with *SSA2* having strong blocking interactions with all time points and *SSA1* mostly

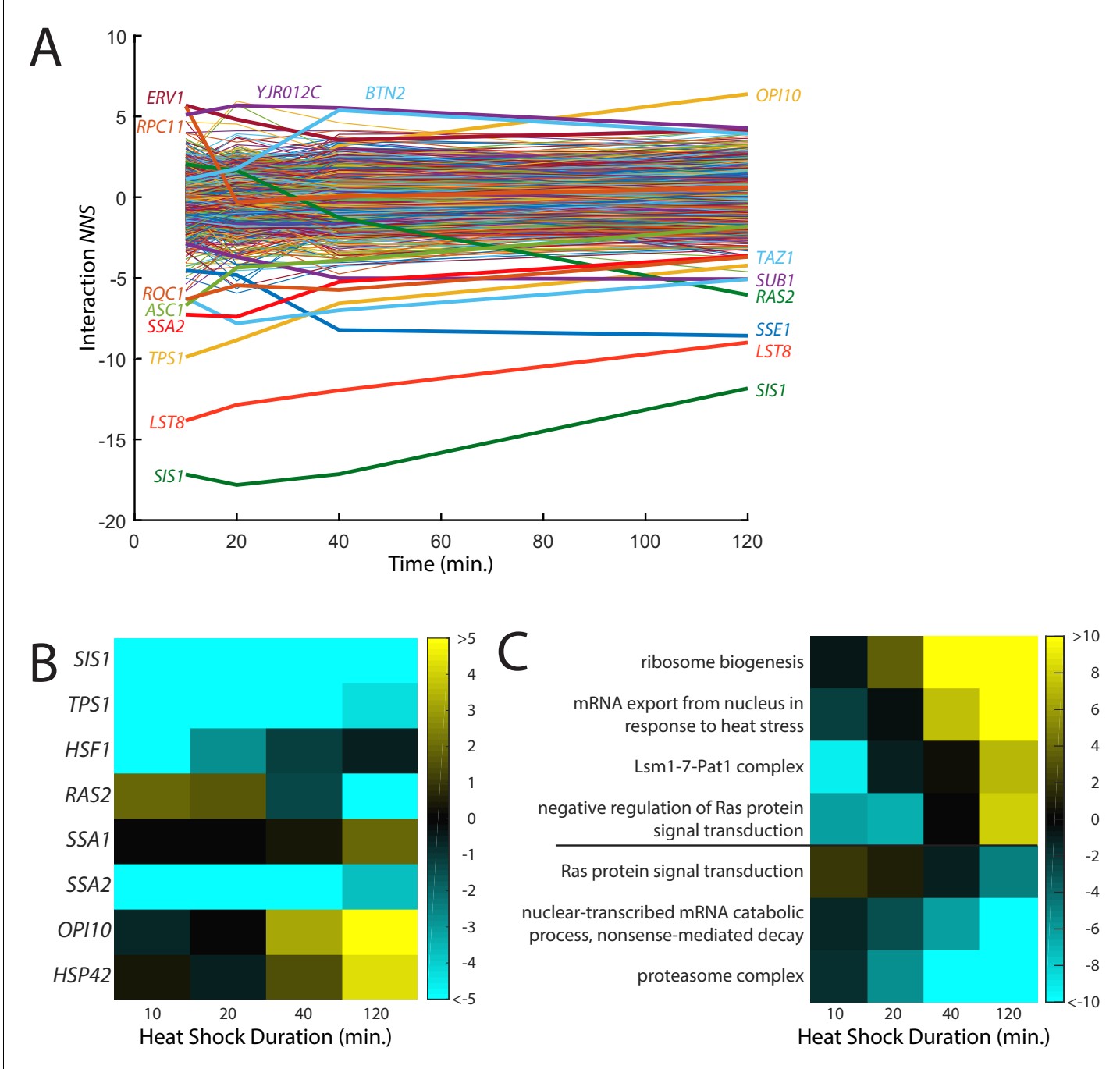

**Figure 4.** Time-resolved interactions between genetic knockdowns and heat stress. (**A**) Time course of interaction *NNS* for each gene with 39°C heat shock. Time points were measured at 10, 20, 40, and 120 min. (**B**) Heat map of interaction *NNS* for selected genes in the screen. (**C**) Heat map of GO category scores for selected categories with differential effects in early and late time points.

The online version of this article includes the following figure supplement(s) for figure 4:

**Figure supplement 1.** Effect of heat shock on CRISPR knockdown efficiency.

not interacting with heat stress but having aggravating interactions at 40 and 120 min. These distinct profiles suggest differential regulation of the HSR by *SSA1* and *SSA2* over prolonged heat stress.

To identify cellular processes that affected the early and late responses to heat shock differently, we analyzed which GO categories had differential effects over the time course (*Figure 4C*;

*Supplementary file 5*). Like the aforementioned GO scores, these GO scores were calculated based on the Student's t-test between the scores of all genes and those of the specific category. Knockdown of genes that are a part of the heat shock response, such as 'mRNA export from nucleus in response to heat stress', generally exacerbated the heat shock response at later time points, suggesting that if the response to heat is impaired, the stress response will be prolonged. Another mRNA regulator, the Lsm1-7-Patl complex, which degrades some mRNAs, also caused a stronger late-phase heat shock response. Additionally, genes involved in Ras signal transduction followed the same trend as *RAS2* and were unusually high at early time points but low at later ones. Conversely, negative regulators of Ras showed the opposite trend. The lowest interaction scores found in later time points were proteasomal genes, while the highest scores were for ribosome biogenesis genes. ReporterSeq can thus identify cellular components that are critical for different phases of a stress response.

## ReporterSeq reveals stress-specific HSR regulators

We next performed ReporterSeq on 11 additional stress conditions in duplicate, including glucose starvation and toxins (*Supplementary file 6*, *Figure 5—figure supplement 1*). These stressors affect cells through a variety of mechanisms and cause varying degrees of HSR reporter activation as measured by qPCR (*Figure 5A*). We applied these stressors for between 20 min and 3 hr. As expected, the pleiotropic proteotoxic stressors and amino acid analogs known to cause accumulation of misfolded protein (*Jacobson et al., 2012*; *Trotter et al., 2002*; *Weids et al., 2016*) robustly activated the HSR. The other stressors, including glucose starvation, and the ER stressors, DTT and tunicamycin, had milder effects on the HSR. We calculated interaction *NNS* for each gene in each stress condition (*Figure 5B*). Some of the stressors shared the same strong genetic interactions that we observed under heat shock conditions. For example, all stressors, with the exception of the proteasome inhibitor bortezomib, had a negative interaction *NNS* with the *SIS1* and *SSA2* knockdowns.

Many genes had strong interactions in some stressors, but not in others. For example, knockdown of the Hsp90 co-chaperone *STI1* (*Chang et al., 1997*; *Nicolet and Craig, 1989*) had aggravating interactions specifically with macbecin and canavanine, two stressors that have been demonstrated to reduce Hsp90 availability (*Alford and Brandman, 2018*), and unexpectedly, bortezomib (*Figure 5C*). Knockdown of *GCN3*, a gene involved in translation initiation regulation (*Yang and Hinnebusch, 1996*), had the strongest aggravating interactions in the screen, hyperinducing the response to hydrogen peroxide, and arsenite stress. Knockdown of *SNF4*, a Snf1 kinase subunit that plays a key role in glucose-repressed gene transcription (*McCartney and Schmidt, 2001*), had a strong aggravating interaction only with glucose starvation. Similar to what we observed in the heat shock time course, we saw distinct interaction profiles from the chaperone genes *SSA1* and *SSA2*, with the *SSA1* knockdown having generally weaker blocking interactions, as well as aggravating interactions with AZC and canavanine, two stressors that were strongly blocked by *SSA2* knockdown. We verified the difference between *SSA1* and *SSA2* interactions with the HSR using gene knockouts and an integrated version of the HSR reporter used in a previous study (*Alford and Brandman, 2018*; *Figure 5—figure supplement 2*). Principal component analysis of chaperone gene *NNS* (chaperone list provided in *Supplementary file 8*) over all stressors revealed that the first two components scored strongest for *SIS1/SSA2* and *SSA1*, respectively (*Figure 5—figure supplement 3A*). This suggests that the distinct interaction profiles of *SIS1/SSA2* and *SSA1* are characteristic of broad classes shared by other genes. The stressors that have the strongest effect on component one are the heat shock conditions and cycloheximide, indicative of the strong interactions these stressors had with *SSA2/SIS1* (*Figure 5—figure supplement 3A, B*). Component 2, on the other hand, was driven most strongly with bortezomib, DTT, canavanine, macbecin, and tunicamycin. Exactly why chaperones like *SSA1* had interactions with these stressors is unclear, but suggests a separate role in stress sensing and response from *SSA2/SIS1*.

To systematically evaluate which classes of genes are important for the HSR under different conditions, we analyzed which GO categories had strong interaction scores with each of the stress conditions (*Figure 5D,E*, GO categories provided in *Supplementary file 7*). We observed a wide diversity of stress-specific interactions. For example, knockdown of translation-related GO categories strongly aggravated the response to ethanol and glucose starvation, and to a lesser extent canavanine, AZC, and DTT. Translation genes had blocking interactions with macbecin and arsenite. Knockdown of mitochondrial genes strongly aggravated response to hydrogen peroxide and

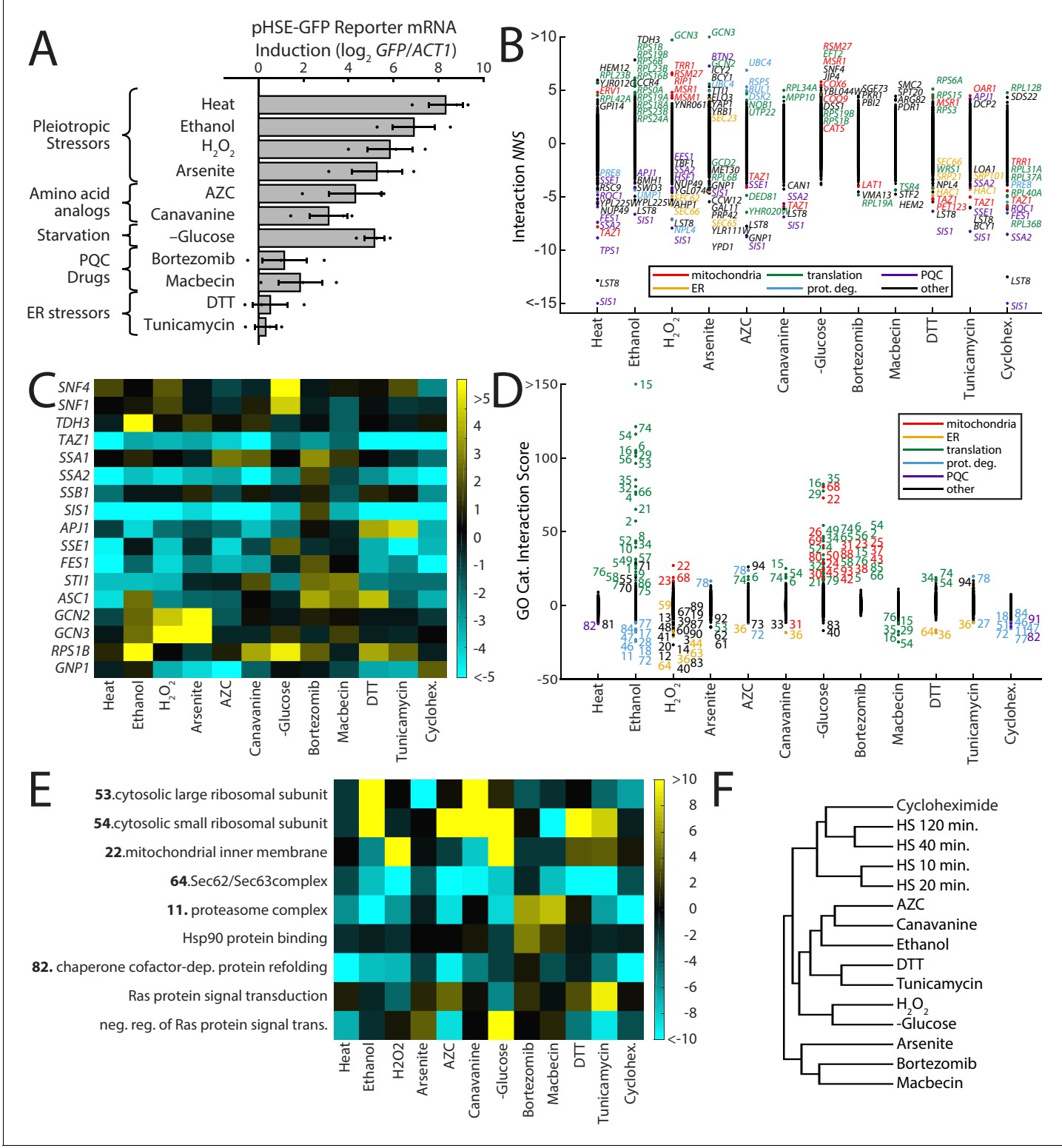

**Figure 5.** ReporterSeq reveals the diversity of responses to different stress conditions. (**A**) HSR reporter mRNA induction based as measured by qPCR in 13 stressors (see **Supplementary file 6** for details of treatments). mRNA levels are relative to *ACT1* mRNA levels, and the results are normalized such that untreated yeast have an induction of 0. Error bars are standard errors of three replicates. (**B**) Interaction *NNS* for each gene under each stressor. Genes with a score magnitude greater than four are labeled (with no more than 12 per condition) and colored based on annotated function. (**C**) Heat map of interaction *NNS* for select genes with each stressor. (**D**) GO category scores for each gene under each stressor. Category ID number key is provided in **Supplementary file 7**. Outlier categories are colored based on function. (**E**) Heat map of GO category scores for selected GO categories

*Figure 5 continued on next page*

*Figure 5 continued*

and each stressor. (F) Hierarchical clustering tree based on gene-stressor interactions. Relatedness between each pair of stressors is quantified by the horizontal length from the branch point.

The online version of this article includes the following figure supplement(s) for figure 5:

**Figure supplement 1.** Replicates of gene–stressors interactions.
**Figure supplement 2.** *SSA1* and *SSA2* have distinct effects on the HSR.
**Figure supplement 3.** Comparisons of proteotoxic stressor effects on the HSR.

glucose starvation. Proteasome knockdown significantly blocked most responses, but had aggravating interactions with macbecin and bortezomib. ER-related genes had blocking interactions with DTT, tunicamycin, hydrogen peroxide, AZC, and canavanine. Enrichment scores for all GO categories are provided in *Supplementary file 5*. The diversity of genetic interactions observed between stressors and the HSR likely reflects effects of the knockdowns on the damage caused by the stressor as well as on stress-specific signaling to the HSR.

A prevalent model for HSR regulation is that the HSR senses changes in chaperone availability, becoming more active when chaperones become less available. In yeast, the Hsp70 and Hsp90 chaperones have both been proposed to have this role. Thus, lowering levels of either of these chaperones should cause constitutive HSR activation and a loss of sensitivity to stressors (i.e. a blocking interaction). Comparing HSR reporter mRNA induction in each stressor to interaction *NNS* for both *SSA2* and *HSC82* (one of two paralogs encoding Hsp90) revealed blocking interaction *NNS* with these chaperones that generally scaled with HSR induction strength, showing that both chaperones were important for responding to stressors (*Figure 5—figure supplement 3C*). Yet some interactions were specific to each chaperone. *HSC82*'s interaction with macbecin was stronger than with AZC, and *SSA2*'s interaction stronger with AZC, consistent with previous results (*Alford and Brandman, 2018*) and reflecting macbecin's inhibition of Hsp90 and *SSA2*'s role in sensing AZC stress.

Notably, cycloheximide had strong blocking interactions with chaperone genes, even though it does not substantially activate the HSR. This may occur through inhibition of translation, which lowers the load of misfolded proteins and requirements for chaperones in the cell and thus may block the HSR induction that occurs when chaperones are downregulated. Strikingly, plotting basal induction vs cycloheximide interactions revealed a negative correlation (correlation coefficient = $-0.28$), similar to the effect seen with heat shock (*Figure 5—figure supplement 3D*). Hierarchical clustering of stressors revealed that cycloheximide clustered with late heat shock (40 and 120 min; *Figure 5F*). Thus, many genetic perturbations that affect the HSR are dependent on translation, consistent with reports that the bulk of misfolded proteins in a cell are newly translated, nascent chains (*Medicherla and Goldberg, 2008*) and the dependence of the HSR on translation in heat stress (*Feder et al., 2021*). Translation-dependent HSR activation may explain blocking interactions observed in other stressors with low HSR induction, such as DTT, tunicamycin, and glucose starvation.

## Gcn3-dependent translation of the HSR reporter

The knockdown with the strongest and most specific stress–gene interactions was the translation initiation factor, *GCN3*, displaying strong aggravating interactions with hydrogen peroxide and arsenite (*Figure 5C*). G is a component of eIF2B, the nucleotide exchange factor for eIF2. *GCN3* has been shown to sense eIF2 phosphorylation and prevent nucleotide exchange of eIF2 thereby halting translation (*Yang and Hinnebusch, 1996*).

To further study the effects of *GCN3* knockdown on the HSR, we used a single CRISPR sgRNA selected from the pool targeting *GCN3*. The HSR reporter encodes GFP, allowing us to measure its output by GFP fluorescence. We measured both GFP mRNA levels and fluorescence in a condition that was predicted to be aggravated by *GCN3* knockdown, arsenite, as well as one that should be unaffected, AZC. *GCN3* knockdown increased mRNA levels of the HSR reporter while reducing its protein levels under arsenite stress (*Figure 6A*), resulting in decreased translation efficiency (*Figure 6B*). As expected, no significant effect of *GCN3* knockdown in both protein and mRNA levels is observed after treatment with AZC (*Figure 6A, B*). This suggests that *GCN3* increases translation efficiency of the HSR reporter under arsenite stress.

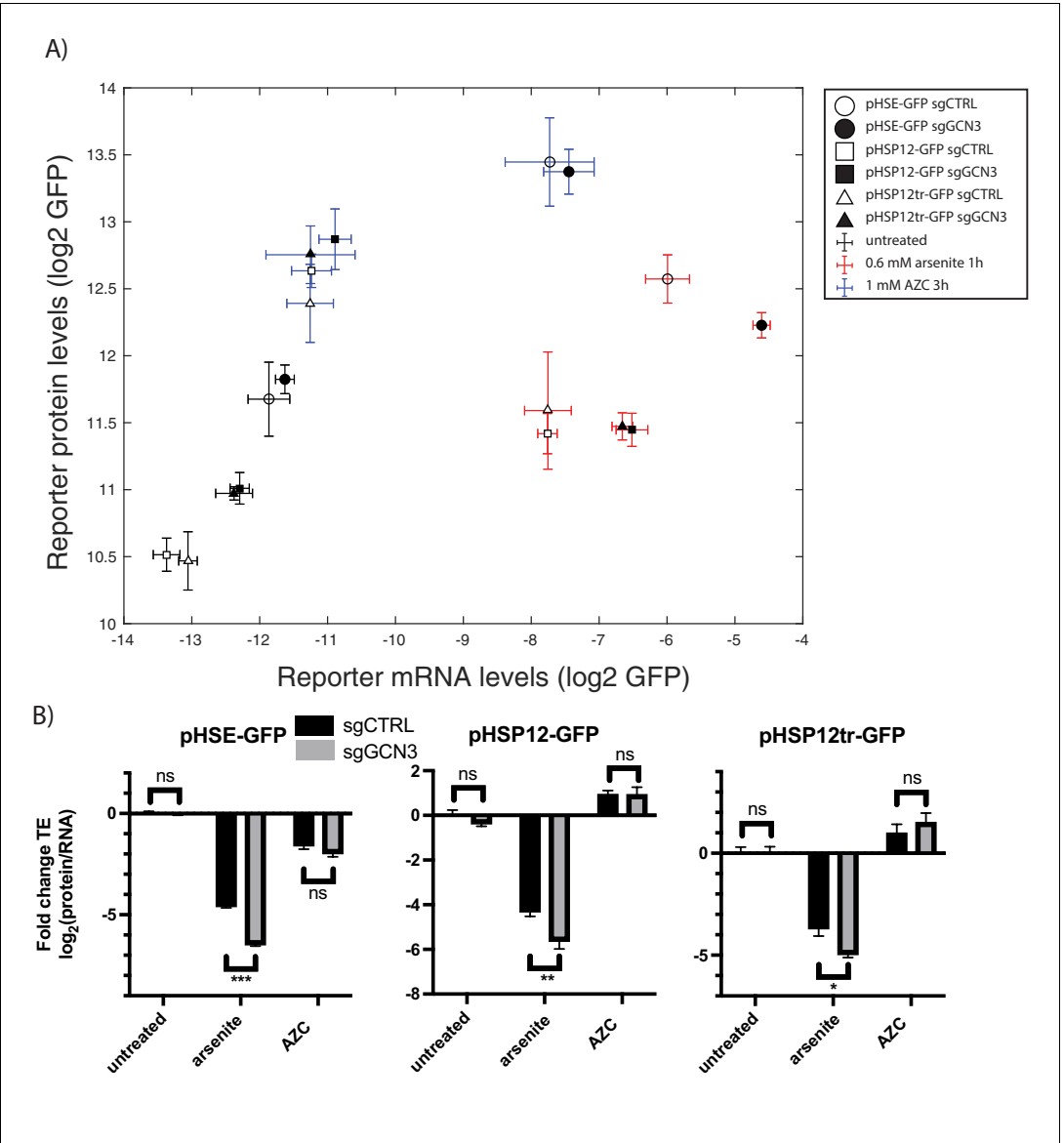

**Figure 6.** GCN3 is required for the efficient translation of stress-induced genes under arsenite stress. (**A**) mRNA levels versus protein levels of the HSR reporter as measured by qPCR and GFP fluorescence with the indicated conditions and sgRNAs. (**B**) Translation efficiency, normalized to control untreated. Error bars are standard errors of six independent biological replicates. sgCTRL targets no gene. p-values are calculated based on an unpaired t-test: [n.s.]p>0.05; *p<0.05; **p<0.01; ***p<0.001.

To determine whether Gcn3-dependent translation in arsenite was a general phenomenon or specific to the HSR reporter, we constructed two additional reporters with identical CDS and 3' UTR as our HSR reporter, but regulated by different stress-dependent promoters. The first reporter includes the full-length promoter sequence of *HSP12*, a stress-responsive gene induced by a variety of stressors that includes an heat shock element (HSE) and seven STRE sequences. In a second reporter, we removed the HSE in this reporter by truncating 805 bp while preserving the STRE sequences. *GCN3* knockdown decreased the translation efficiency of both versions of the *pHSP12* reporters under arsenite stress (***Figure 6B***), while translation efficiency remained unchanged during AZC treatment (***Figure 6B***). No effects of *GCN3* knockdown on translation efficiency were observed for any reporters in unstressed conditions (***Figure 6B***). These findings suggest a general role for *GCN3* in increasing translation efficiency of stress-responsive genes under arsenite stress.

## Discussion

We dissected the genetic modulators of the HSR using ReporterSeq, a pooled, high-throughput screening method that measures the effect of genome-wide genetic perturbations on the expression of a single-reporter-driven RNA. ReporterSeq couples gene expression to levels of barcodes that can be read with deep sequencing, thus enabling pooled, full-genome screening without the need for cell enrichment (e.g. via fluorescence-based cell sorting). This allowed us to identify known and novel regulators of the HSR and measure their roles in responding to diverse stressors in a time-resolved manner. Because ReporterSeq measures mRNA levels, we were able to observe modulators with differential effect on mRNA and protein levels, such as *GCN3*, which we discovered increases translation efficiency of stress-responsive genes in some stress conditions.

The strongest HSR-regulating pathway identified by our screen was the Hsp70/Hsp40 pair, *SSA2/SIS1*. Knockdown of *SIS1* or *SSA2* desensitized cells to every stressor except for the proteasome inhibitor bortezomib, a weak activator of the HSR. This is consistent with studies demonstrating direct binding between Hsp70 and Hsf1 (*Zheng et al., 2016*) and suggesting that Sis1 is the primary Hsp40 facilitating this interaction (*Feder et al., 2021*). We identified one strong regulator of the HSR, Asc1, as an HSR regulator independent of Ssa2/Sis1. We note that the mechanisms through which genetic perturbations affect the HSR can vary in directness, from reducing levels of an Hsf1-binding protein to increasing levels of a toxic intermediate. Like other genetic screening methods, ReporterSeq cannot distinguish between these cases. Future studies may allow a more systematic delineation of the independence and mechanisms of pathways driving the HSR.

Both the Hsp90 inhibitor, macbecin, and the proteasome inhibitor, bortezomib, had somewhat similar global gene interaction profiles that differed from other stressors. They both featured aggravating interactions with the proteasome, Hsp90 binding genes, and *SSA1*. Furthermore, bortezomib was the only stressor to positively interact with *SIS1* and *SSA2*, while macbecin had little or no interaction with these genes. This suggests that macbecin and bortezomib may activate the HSR via a separate mechanism from Ssa2/Sis1, and is consistent with a previous study proposing separate activation mechanisms of the HSR by Hsp70 (Ssa2) and Hsp90 (*Alford and Brandman, 2018*).

The mechanisms by which many of the gene and gene categories we identified regulate the HSR are at this moment unknown. For example, a major mystery is how genes regulating translation and mitochondria, which showed strong interactions with multiple conditions, regulate the HSR. Additionally, how the Ras/cAMP pathway and Taz1, which has been proposed to be essential for protein homeostasis (*de Taffin de Tilques et al., 2018*), regulate the HSR is poorly understood. Interaction data from our ReporterSeq experiments may be useful in formulating hypotheses that can be tested with detailed follow-up experiments.

We identified a novel regulator of the HSR, *GCN3*, under specific stress conditions, such as treatments with sodium arsenite, ethanol, or hydrogen peroxide. *GCN3* encodes the α subunit of the eIF2B complex, the guanine nucleotide exchange factor for eIF2. Under a variety of stress conditions, the eIF2 kinase, Gcn2, phosphorylates eIF2α causing inhibition of eIF2B activity through interactions between eIF2B–eIF2α-phosphorylated complex (*Marintchev and Ito, 2020*). This results in global translation repression while inducing the integrated stress response (ISR) through translation of *GCN4* (*Yang and Hinnebusch, 1996*). *GCN2* shows a similar interaction with the same stressors as *GCN3* (*Figure 5C*), *consistent with a role for the eIF2B–eIF2α complex in regulating the HSR*. Gcn3 is essential for regulating translation under stress, as eIF2B activity is resistant to inhibition by phosphorylated eIF2α when *GCN3* is deleted (*Kimball et al., 1998*; *Yang and Hinnebusch, 1996*). Thus, in stress conditions that promote eIF2α phosphorylation, *GCN3* would be expected to lower translation efficiency of non-ISR-related genes such as our reporter genes. Yet our data suggest a role for *GCN3* in increasing their translation efficiency after treatment with sodium arsenite. This effect is not limited to genes that are Hsf1 dependent, since we observed similar results with reporter genes regulated by two variants of the *HSP12* promoter (*Chowdhary et al., 2019*). These observations shed light on new roles of *GCN3* in increasing translation efficiency of stress-responsive genes under stress independently of the ISR. Further characterization is required in order to determine how *GCN3* affects the translation efficiency of non-stress induced genes under stress conditions.

Our implementation of ReporterSeq allows full-genome screens to be completed in a single-cell culture and multiple screens can be read out using a single sequencing run. We expect ReporterSeq

results to improve with greater depth of sequencing and sample size. Yet even with the relatively modest level of sequencing in this study (~5 M–15 M reads per sample; *Supplementary file 9*), we identified myriad stress-specific regulators of the HSR, including known and novel regulators, creating a comprehensive genetic view of a major protein quality control pathway. The scale and precision of ReporterSeq is only limited by growth conditions and DNA sequencing, not cell enrichment, microfluidics, or cell processing. Multiple libraries can be multiplexed in pooled growth conditions. A comparison of ReporterSeq with a previous screen for HSR regulators revealed a weak correlation for basal HSR activity and enriched stress–gene interactions for hits in the previous screen (*Figure 2—figure supplement 3*), demonstrating that ReporterSeq can reveal features of known novel regulators. ReporterSeq is ideally suited to study many stress pathways, like the HSR, which have been difficult to comprehensively evaluate because they are co-regulated with translation and thus fluorescence- and luciferase-based assays that require translation are limited in effectiveness. Indeed, contemporaneous to this work, a similar technique was used to dissect the genetic regulators of Gcn4 (*Muller et al., 2020*), regulator of the yeast general amino acid control response, a stress-response homologous to the integrated stress response in metazoans. Applied broadly, ReporterSeq may significantly advance the study of cellular pathways in vivo.

## Materials and methods

### Yeast strains and growth conditions

All experiments were performed with BY4741 yeast. The *ssa2* knockout (yJP385) was made using *NAT* replacement of the *SSA2* ORF and selected with nourseothricin. Yeast were grown at 30℃ (unless otherwise indicated) in synthetic defined (SD) media with the appropriate dropout. SD media used for growth of yeast cultures contained: 2% w/v dextrose (Thermo Fisher Scientific, Waltham, MA), 13.4 g/L Yeast Nitrogen Base without Amino Acids (BD Biosciences, San Jose, CA), 0.03 g/L L-isoleucine (Sigma-Aldrich, St. Louis, MO), 0.15 g/L L-valine (Sigma-Aldrich), 0.04 g/L adenine hemisulfate (Sigma-Aldrich), 0.02 g/L L-arginine (Sigma-Aldrich), 0.03 g/L L-lysine (Sigma-Aldrich), 0.05 g/L L-phenylalanine (Sigma-Aldrich), 0.2 g/L L-threonine (Sigma-Aldrich), 0.03 g/L L-tyrosine (Sigma-Aldrich), 0.018 g/L L-histidine (Sigma-Aldrich), 0.09 g/L L-leucine (Sigma-Aldrich), 0.018 g/L L-methionine (Sigma-Aldrich), 0.036 g/L L-tryptophan (Sigma-Aldrich), and 0.018 g/L uracil (Sigma-Aldrich).

### Plasmid library construction

Approximately 12 sgRNAs were designed to target the 5' end of each protein-coding and non-coding gene in the genome (between −750 and +50 nucleotides from the start of the ORF with priority toward sgRNAs closer to the ORF start) based on the Saccharomyces Genome Database (*Cherry et al., 2012*). A pool of oligonucleotides was ordered (CustomArray, Inc, Bothell, WA) containing these sequences and constant flanking sequences for the purposes of cloning (*Supplementary file 10*). A 2μ -URA plasmid was constructed from three DNA fragments using HiFi DNA assembly Master Mix (New England Biolabs, Ipswich, MA) for cloning. The first fragment was a diverse sample of inserts amplified via PCR using KAPA HiFi polymerase (Roche Basel, Switzerland), producing a sequence that contained random barcodes associated with a random sgRNAs from the library. This first fragment was generated in three steps. Step (1a) generation of a constant region using the parental plasmid as template, (1b) generation of random sgRNA sequences using the custom pool of oligonucleotides containing the sgRNA sequences, with flanking sequence to that of product of step 1a in its 5' end. Step (2) PCR product using products of steps 1a and 1b to generate a constant region containing the random sgRNAs. Step (3) addition of random barcodes via PCR using product of step two as template, generating random barcodes in the 5' end – which corresponds to the 3' UTR of GFP – associated with random sgRNA sequences. The second fragment was the parent plasmid digested at two locations using the FastDigest enzymes BglI and BglII (Thermo Fisher, Sunnyvale, CA). The final fragment was another region of the plasmid amplified via PCR, which was necessary to connect the previous two pieces. These three fragments were assembled in a 100 µL reaction containing 700 ng of the first fragment, 600 ng of the second fragment, and 400 ng of the final fragment. After a 1 hr incubation at 50℃, the reaction was concentrated to 20 µL using a DNA clean and concentrate column (Zymo Research, Irvine, CA) and electroporated in five separate cuvettes containing 4 µL of the plasmid and 40 µL of ElectroTen-Blue electrocompetent

cells (Agilent, Santa Clara, CA). After a 1 hr outgrowth in LB at 37°C, the plasmid was then grown in 1 L of LB containing carbenicillin for 2–3 days until it reached saturation. The plasmid was then purified via maxiprep (Qiagen, Hilden, Germany). A diagram of the plasmid library construction strategy and sequences used is shown in supplemental data files library_creation_diagram, library_primer_sequences, and library_sequences found in *Supplementary file 11*.

## Plasmid library sequencing

Two micrograms of plasmid was cut with the FastDigest PstI restriction enzyme (Thermo Fisher), and the 900 bp piece was gel purified using a gel DNA extraction kit (Zymo). This piece was then recircularized in a 20 µL reaction containing 60 ng of gel purified DNA and 1 µL T4 DNA Ligase (Thermo Fisher) in buffer. Cutting and recircularizing was necessary to increase efficiency of later sequencing steps for unknown reasons. The ligation product was then amplified with primers containing the appropriate overhangs for high-throughput sequencing and then gel purified. This piece was then sequenced through paired-end sequencing using custom primers on an Illumina (San Diego, CA) sequencing machine (Miseq or Nextseq).

## Yeast library construction

Yeast were first transformed with a 2-micron pMET17-dCas9-MxiI/*LEU2* plasmid (adapted from *Gilbert et al., 2013*) using standard methods. Five milliliters of this yeast was grown overnight in SD -Leu media to saturation. This yeast was then back diluted into a 1 L YPD culture to approximately ~0.1 $OD_{600}$. This culture was then grown at 30°C with shaking until the $OD_{600}$ reached ~0.6. The yeast were transformed with 600 ng of the library plasmid using standard procedures. However, instead of plating the yeast after the heat shock, they were grown in 1 L of SD -Ura -Leu for 3 days. The yeast were then either rediluted for experiments or frozen in 1 mL aliquots in media with 30% glycerol at −80°C for later use.

## Yeast sample preparation and screening

Yeast from a saturated library sample were grown overnight to log phase in SD -Ura/Leu. These yeast were then split into a number of cultures of at least 10 mL. Yeast were then either left untreated or stressed under the various conditions for the indicated times (*Supplementary file 6*). The yeast were then vacuum filtered onto nitrocellulose paper and flash frozen in liquid nitrogen. All replicates (see *Supplementary file 1*) are biological replicates, which came from separate transformations of the library into the yeast, separate treatments, and sample preparations. Some basal samples included technical replicates (independent sequencing sample preparation of the same yeast sample), and these were averaged into one biological replicate.

## Nucleic acid extraction and sequencing sample preparation

RNA was extracted through acid–phenol extraction. The frozen yeast pellets were first resuspended in 2.7 mL AE Buffer (50 mM sodium acetate pH 5.5, 10 mM EDTA), 0.3 mL 10% SDS, and 3 mL acid phenol:chloroform (Thermo Fisher). This mixture was heated to 65°C and shaken at 1400 rpm for 10 min followed by 5 min on ice. The solution was then spun at 13,000 g for 15 min. The supernatant was then added to 3 mL of chloroform and spun at 15,000 g for 5 min. The supernatant was then purified through a standard isopropanol extraction.

We then used NextFlex oligo-dT beads (PerkinElmer, Waltham, MA) to select polyadenylated mRNA from 100 µg of the extracted RNA. This sample was then subjected to DNAse treatment using Turbo DNase (Thermo Fisher) treatment and then purified using AMPure XP beads (Beckman Coulter, Indianapolis, IN). The resulting RNA was reverse transcribed using Multiscribe reverse transcriptase and a gene-specific primer (Thermo Fisher), and the cDNA was purified using alkaline lysis of the RNA followed by another round of AMPure bead purification.

To obtain the yeast DNA samples, frozen yeast was thawed and miniprepped using a Zymoprep Yeast Plasmid Miniprep Kit (Zymo Research).

The purified cDNA and the purified DNA were then amplified at the barcode region using primers with the appropriate Illumina sequencing ends and indexing barcodes to differentiate each sample. These resulting PCR products were then gel purified, mixed proportionately with each other, and then sequenced using a Illumina next-generation sequencing machine (MiSeq, NextSeq,

or NovaSeq). The total number of barcode reads matching sgRNAs for each condition is listed in *Supplementary file 9*.

## Scoring/statistical analysis

Basal neighborhood normalized scores (**NNS**) and interaction *NNS* were analyzed as pairs of conditions. For the case of the basal *NNS*, RNA levels and DNA levels were compared, whereas for the gene-stressors interaction *NNS*, stressor RNA was compared to unstressed RNA. While choosing the sgRNAs for library construction, an sgRNA mapped to two adjacent genes if the sgRNA targeted within 750 nucleotides upstream of both ORFs. For each pair of conditions, $A$ and $B$, and each barcode, $x$, we assign an *NNS*, $score(x, A, B)$, which captures how extreme (positive or negative) the read count is for barcode $x$ in condition $A$ versus condition $B$. These scores allow us to compare, in a principled fashion, the discrepancies between counts in different conditions, enabling a convenient aggregation to form a score for each gene or gene cluster. Crucially, the scores take into account (1) the higher relative amount of noise in small counts versus larger counts and (2) the fact that some conditions yield higher read counts across most barcodes.

We now describe and motivate our definition of $score(x, A, B)$. Let $A(x)$ denote the logarithm of the count of barcode $x$ under condition $A$, and let $B(x)$ denote the logarithms of the counts for barcode $x$ under conditions $B$, scaled by a factor $c_{A,B}$ that compensates for the possibility that some conditions yield systematically higher counts than others. After this scaling, for most barcodes, $x'$, it is the case that $A(x') \approx B(x')$.

Let $d_{A,B}(x) = \frac{A(x) - B(x)}{A(x) + B(x)}$ denote the normalized discrepancy in these values, and note that this quantity is bounded between $\pm 1$. The score is this discrepancy, $d_{A,B}(x)$ after the median value of this is subtracted, normalized by a factor $S_{A(x),B(x)}$ that takes into account how much variance is typical in this quantity, given the values of $A(x)$ and $B(x)$. Note that we expect $d_{A,B}(x)$ to have a larger variance due to random noise in the experiment and measurement when $A(x)$ and $B(x)$ correspond to small counts. This scaling factor $S_{A(x),B(x)}$ is defined to be an outlier-robust analog of the standard deviation of $d_{A,B}(x')$ computed across the set of barcodes $x'$ for which $A(x') + B(x') \approx A(x) + B(x)$.

The following pseudocode formalizes the above calculation of the *NNS* for each barcode in each pair of conditions:

Calculate barcode: *NNS*
Input; Barcode *x*, read counts for all barcodes in conditions *A* and *B,* and scaling factor *c(A,B).*
Output: *score(x,A,B)*

- Define parameters Cohort_Size = 10,000 and Clip_Param = 100.
- Let *A*(x) denote the logarithm of 1 + *read_count* for barcode *x* in condition *A*, and let *B*(x) denote the logarithm of 1 + *read_count* for barcode *x* in condition *B* scaled by a constant *c(A, B)* defined as the ratio of the median of the logarithms of the top 1000 counts in condition *A*, to the median of the logarithms of the top 1000 counts in condition *B*.
- Define the set of barcodes

$$X = x' : |A(x') + B(x') - A(x) + B(x)| < \text{delta},$$

where delta is the smallest value such that $|X| \geq$ Cohort_Size.

- For each *x'* in *X*, let *d(x') = (A(x)-B(x))/(A(x)+B(x))*
- Define *S* = truncatedStandardDeviation(d(X),Clip_Param), which is the standard deviation of the multiset of values of d(x') taken across all elements x' of X, after the largest and smallest Clip_Param values are removed. (E.g. if Clip_Param = 100, the largest and smallest 100 values are removed prior to computing the standard deviation.)
- Return *score(x,A,B)=[d(x)-median(d(X))]/S*.
- If the sum of the read counts in conditions A and B is less than 10, we discard that score, and do not use it when computing gene scores of GO scores.

Basal *NNS* (four replicates) and stress interaction *NNS* (two replicates) were computed as the arithmetic mean of replicates.

GO scores were calculated based on the above calculated *NNS* after saturation of the top and bottom 0.5% *NNS* (to prevent strong outliers from dominating a category score). For each GO

category, the distribution of *NNS* for genes within that category and those of the entire distribution (scores for all genes) were compared using a Student's t-test. The GO score was the natural log of the reciprocal of the resultant p-value from the Student's t-test. The GO score sign indicates in whether the mean of the gene scores for that gene category was above or below zero.

Hierarchical clustering was performed using Cluster 3.0 (*de Hoon et al., 2004*), using uncentered correlation similarity metric and complete linkage clustering method.

Mitochondrially encoded genes were removed from supplementary data file one and subsequent analysis. Clustering and GO enrichment was performed on all genes except tRNAs and retrotransposons. Excluded gene list is provided in *Supplementary file 12*.

Fold changes for each gene were by averaging the $\log_2$ fold changes for each barcode count and subtracting the median fold change of the distribution. Counts less than five were excluded.

MATLAB computer programs and raw data to compute gene scores and GO category enrichment are provided in *Supplementary file 12*.

## Flow cytometry

Yeast were grown overnight from a saturated culture to log phase (OD < 0.6), such that they were growing in log phase for at least 14 hr. These yeast were then measured on an Accuri flow cytometer at 10,000 yeast per sample, under the fast flow setting. Each flow cytometry data point is a biological replicate, in which the steps starting from the wild-type (or in some cases knockout) yeast strain are repeated separately for each replicate, including plasmid transformations and treatment conditions.

## Quantitative PCR

RNA was extracted, and treated with DNase according to the protocol described above. One microgram of RNA was then reverse transcribed using Multiscribe reverse transcriptase (Thermo Fisher) and random hexamers in a 25 µL reaction. One microliter of the cDNA was then mixed with Luna qPCR master mix (New England Biolabs) and primers (to a final concentration 100 nM), which amplified either GFP or *ACT1*, the reference gene. These mixtures were measured in triplicate for each sample-primer combination with a ViiA 7 qPCR machine (Thermo Fisher), and the median $C_T$ value was used in analysis. Each qPCR data point is a biological replicate, in which the steps starting from the wild-type yeast strain are repeated separately for each replicate, including plasmid transformations and treatment conditions.

## Acknowledgements

We thank L Persson, C Sitron, J Giafaglione, J Park, S Alfonso, Drs. Z Davis, Z Jaafar, R Rohatgi, M Krasnow, J Weissman, L Gilbert, and C Jan for helpful suggestions throughout the project and in preparation of the manuscript. Funding was provided by the Lucille P. Markey Basic Biomedical Research Fellowship and NIH 5 T32 GM007276 to BDA, and NSF Awards 1813049 and 1704417 and ONR YIP award N00014-18-1-2295 to GV, and R01GM115968 to OB.

## Additional information

### Funding

| Funder | Grant reference number | Author |
| --- | --- | --- |
| National Institutes of Health | R01GM115968 | Onn Brandman |
| National Institutes of Health | 5 T32 GM007276 | Brian D Alford |
| National Science Foundation | 1813049 | Gregory Valiant |
| National Science Foundation | 1704417 | Gregory Valiant |
| Office of Naval Research | N00014-18-1-2295 | Gregory Valiant |

The funders had no role in study design, data collection and interpretation, or the decision to submit the work for publication.

## Author contributions
Brian D Alford, Conceptualization, Data curation, Software, Formal analysis, Investigation, Visualization, Methodology, Writing - original draft, Writing - review and editing; Eduardo Tassoni-Tsuchida, Conceptualization, Investigation, Methodology, Writing - original draft, Writing - review and editing; Danish Khan, Jeremy J Work, Methodology; Gregory Valiant, Conceptualization, Data curation, Software, Formal analysis, Methodology, Writing - review and editing; Onn Brandman, Conceptualization, Resources, Data curation, Software, Formal analysis, Supervision, Funding acquisition, Validation, Investigation, Visualization, Methodology, Writing - original draft, Project administration, Writing - review and editing

## Author ORCIDs
Eduardo Tassoni-Tsuchida (ID) https://orcid.org/0000-0001-8062-6027
Jeremy J Work (ID) https://orcid.org/0000-0002-1677-4954
Onn Brandman (ID) https://orcid.org/0000-0002-2084-154X

## Decision letter and Author response
Decision letter https://doi.org/10.7554/eLife.57376.sa1
Author response https://doi.org/10.7554/eLife.57376.sa2

## Additional files

### Supplementary files
• Supplementary file 1. Basal gene *NNS* for and gene–stress interaction *NNS* for all stressors used in this study.

• Supplementary file 2. Fold change analysis of the top and bottom 20 basal *NNS* and stress interaction *NNS* genes identified in this study (average fold change for all barcodes).

• Supplementary file 3. *NNS* for individual sgRNAs (average of all barcodes) for all genes of this study.

• Supplementary file 4. Individual sgRNA counts and *NNS* for all the genes in every stress condition used in this study.

• Supplementary file 5. Gene ontology (GO) category scores for basal and stress conditions.

• Supplementary file 6. Description of stressors used in this study.

• Supplementary file 7. ID-value key for gene ontology (GO) categories used in *Figure 5D*.

• Supplementary file 8. List of chaperones used in principle components analysis of *Figure 5—figure supplement 2*.

• Supplementary file 9. Total read counts matching barcodes for all conditions used in this study.

• Supplementary file 10. sgRNAs sequences used in the CRISPR library used in this study.

• Supplementary file 11. Library design strategy with primer and library sequences.

• Supplementary file 12. Zip file containing raw read count data, matlab code to generate scores, and library construction schematic and sequences.

• Transparent reporting form

### Data availability
Full datasets are provided in supplementary materials.

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
