## [Decision Letter]

Thank you for submitting your article "Genome-wide, time-sensitive interrogation of the heat shock response under diverse stressors via ReporterSeq" for consideration by *eLife*. Your article has been reviewed by 2 peer reviewers, and the evaluation has been overseen by a Reviewing Editor and Naama Barkai as the Senior Editor. The following individuals involved in review of your submission have agreed to reveal their identity: Allan Drummond (Reviewer #1); Lars Steinmetz (Reviewer #2).

The reviewers have discussed the reviews with one another and the Reviewing Editor has drafted this decision to help you prepare a revised submission.

As you will see below, both reviewers were excited about the screening tool you introduced and were supportive about the work. The main concern was the lack of sufficient validation. The revision would therefore need to include all validation of key claims. In addition please address also all comments below.

*Reviewer #1:*

In this paper, Alford et al., introduce a clever and obviously useful genetic screening tool termed ReporterSeq, and use it to identify regulators of various stress responses in budding yeast. The heat shock response (HSR) is a conserved transcriptional response to a variety of stresses driven primarily by the transcriptional activator Hsf1. Recent work has established that Hsf1 is primarily regulated by chaperones (primarily Hsp70 with a possible contribution of Hsp90) binding to and inhibiting Hsf1. During stress, this inhibition is relieved by titration of the chaperone off of Hsf1, allowing for induction of Hsf1 target genes. While this basic mechanism is well understood, it remains an important question how diverse stresses lead to shared activation of Hsf1 and how these different pathways might be regulated.

This study investigates that question by doing a number of genetic screens for genes which regulate the HSR in different stress conditions. This multiplicity of screens is aided by the ReporterSeq technology, which combines a CRISPRi screen with a promoter- specific readout of gene induction. The authors identify a number of genes which differentially regulate the HSR during a variety of stresses and offer some hypotheses for pathway specific effects. They propose that ASC1 depletion, bortezomib and macbecin all activate the HSR through a separate mechanism from the Hsp70 dependent pathway described. They also identify a link between GCN3 and translation regulation of their HSE containing reporter.

Taken by itself, ReportSeq represents a clear and important step forward in genetic screening methodologies, combining the ease and genetic specificity of CRISPRi screens with a more nuanced phenotypic readout than growth. The limitations of the method should be more clearly spelled out. Some technical details of the implementation could use more exploration, such as the effect of multi-day activation of gene repression and the effectiveness of various sgRNAs, but the authors clearly demonstrate the usefulness of the method. The novel insights into the HSR, however, are less convincing. Similar screens have been carried out by Onn Brandman using a fluorescence based reporter and synthetic gene arrays (Brandman et al., Cell 2012) and the findings are largely in agreement. The broader significance of particular cases highlighted in the paper would benefit greatly from validation beyond the synthetic reporter used in this study. In addition, the interaction scores used to quantify the effects of gene knockdown are influenced by the extent to which each gene regulates the basal HSR, which complicates the interpretation of some of the results.

1. In the introduction, "Additionally, post-transcriptional models have been proposed, such as HSR-regulated mRNA half life (Heikkinen, 2003) or regulation of the translation efficiency of heat shock mRNAs (Zid and O'Shea, 2014). Yet the extent to which these and other mechanisms drive the HSR under diverse stressors is poorly understood…" This section introduces the HSR as a transcriptional response, and states that it is driven by Hsf1, but mRNA half life or translational efficiency are not thought to regulate or activate Hsf1 or the transcriptional response; these are downstream of the HSR. Later material in the manuscript indicates that "regulation of the HSR" includes translation, not merely transcription as the introduction states. The writing should be clarified.

2. Please provide more detail on the reporter construct used. It is suggested that it is a combination of HSEs and the CYC1 promoter as used in Brandman et al., 2012. However the authors say only that it is "similar" to that reporter without specifying the ways in which it is different. Some additional discussion of the extent to which this reporter is regulated by Hsf1 vs other stress-induced transcription factors such as Msn2/4 is important for interpreting the results of the paper. For instance, any Hsp70-independent regulation could either be a result of Hsp70-independent regulation of Hsf1 or non-Hsf1 regulation of the reporter. I recommend additional experiments be done combining CRISPRi knockdown of key genes such as ASC1 and HSC82 with additional reporter constructs, either with promoters known to be regulated by Hsf1 or Msn2/4 or with other synthetic promoters containing STRE elements.

3. The analysis would benefit from a multiple-testing correction. What the authors call "confidence" is effectively statistical significance of (replicated) scores under a normal null model, a Z score, which has an associated probability. We urge the authors to apply a false discovery rate (FDR) correction, or their choice of multiple-testing correction, to assess the adjusted probability that the observed scores occur by chance in a dataset of this size.

4. The induction of the reporter after knockout of gene X is interpreted as meaning that X regulates the HSR. However, knocking out X could (say) cause build-up of a toxic intermediate in some pathway, triggering the cellular stress response and thus activation of the HSR. X here would not be a "regulator" of the transcriptional response in the sense in which the authors intend. The logic of the connection between signal and interpretation should be more clearly spelled out and its limitations or edge cases mentioned.

5. More technical details in the Methods section would be valuable to the community. For instance a diagram of the cloning process would be useful. In addition, providing the sequences of the adapters and primers and the times and concentrations of enzymes used in library preparation would allow for replication of the technique.

6. The inclusion of the code used and the pseudocode description of it is greatly appreciated. One question that remained is how the biological replicates are used in the creation of the score. The only time that the replicates are used or shown are in the volcano plots in Figure 2C. Supplemental figures showing the reproducibility across replicates or a description of how the replicates were used in the scoring process would be helpful.

7. In Figure S1A no induction of the reporter is seen with a SSA2 targeting sgRNA. This is at odds with subsequent data showing a strong effect of SSA2 knockdown. Presumably, this is a result of poor efficacy of the chosen sgRNA and additional sgRNAs should be tested to resolve this discrepancy. Furthermore, it would be useful to know the extent of knockdown, for instance by tracking SSA2 mRNA levels by qPCR.

8. Later, the authors point out the importance of having multiple guide RNAs and barcodes for each gene. It would be helpful to see how important these factors are by splitting out the results for a select set of genes (perhaps CYC1, SIS1 and SSA2) into guide RNA- and barcode- specific effects.

9. One drawback to the approach used here is that the library is grown for multiple days with active CRISPRi targeting of genes. Presumably, this could lead to the depletion of essential genes from the pool and a discussion of the extent to which this depletion was observed would provide important context for the breadth of the screen.

10. The authors mention that sgRNAs targeting the reporter promoter (CYC1 sgRNAs) do not prevent its activation during heat stress. This implies that knockdown under basal conditions does not necessarily prevent the induction of genes under heat shock conditions. This has important ramifications for the interpretation of later time points of the heat shock time course, partly in regards to heat shock induced genes, as the differential effects seen could result from changes in the extent of the knockdown. Western blots showing the extent of depletion for a subset of genes (such as SSA2, SSA1, SIS1 etc.) with selected guide RNAs during the time course would alleviate this concern.

11. The identification of differences in the roles of SSA1 and SSA2, and Sis1 as opposed to Ydj1 are both important contributions. However, in order to demonstrate approve a difference, it is necessary to show that these genes are all being knocked down to a similar extent. For example, SSA2 is induced by heat shock far less than SSA1, and the authors observe a time-dependent effect of SSA1 but not SSA2, raising the question as to what extent SSA1's induction profile has been altered by knockdown. Thus weI suggest additional experiments showing differential effects on HSR induction using specific guide RNAs with western blots or qPCR to show the extent of depletion. Alternatively the authors could note that they have no evidence for the knockdown of these genes during the time-course or for the relative levels of the knockdowns.

12. Because the interaction score used is based on the basal levels of regulation, genes which strongly affect the basal response such as Sis1 are expected to have strong negative interaction scores. Thus it seems that both the plots shown in Figure 3C and the PCA analysis in Figure S2 could be largely explained by the basal score of genes such as SIS1, SSA2 and SSE1. Some more discussion of how to disentangle genetic interactions with stress from the effects of knockdown on the basal HSR would be very helpful for interpreting these results.

13. Figure 6 does not convincingly show that GCN3 knockdown selectively affects translation of the reporter as opposed to a more general effect. Additional experiments as in Figure 6B but with reporters containing different promoters and 5' UTRs would help prove that the reporter is unique in its response to GCN3 knockdown.

*Reviewer #2:*

The authors introduce ReporterSeq, a new method for high-throughput sequencing-based readout of a reporter gene, and apply it to uncover regulation of the heat shock response. Although this response has been characterized i.e. on transcriptome and proteome changes, our understanding on the regulation of this still limited. The study extends previous work from Brandman et al., (2012) by refining perturbation and readout. ReporterSeq measures DNA and RNA barcodes, or only RNA barcodes, and is combined with genome-wide CRISPR interference. The presented study applies this technique to yield unparalleled insights into Hsf1 regulation in different stresses and over time. I recommend publication of this nice study and have no major concerns.

---

## [Author Response]

Reviewer #1:In this paper, Alford et al., introduce a clever and obviously useful genetic screening tool termed ReporterSeq, and use it to identify regulators of various stress responses in budding yeast. The heat shock response (HSR) is a conserved transcriptional response to a variety of stresses driven primarily by the transcriptional activator Hsf1. Recent work has established that Hsf1 is primarily regulated by chaperones (primarily Hsp70 with a possible contribution of Hsp90) binding to and inhibiting Hsf1. During stress, this inhibition is relieved by titration of the chaperone off of Hsf1, allowing for induction of Hsf1 target genes. While this basic mechanism is well understood, it remains an important question how diverse stresses lead to shared activation of Hsf1 and how these different pathways might be regulated.This study investigates that question by doing a number of genetic screens for genes which regulate the HSR in different stress conditions. This multiplicity of screens is aided by the ReporterSeq technology, which combines a CRISPRi screen with a promoter- specific readout of gene induction. The authors identify a number of genes which differentially regulate the HSR during a variety of stresses and offer some hypotheses for pathway specific effects. They propose that ASC1 depletion, bortezomib and macbecin all activate the HSR through a separate mechanism from the Hsp70 dependent pathway described. They also identify a link between GCN3 and translation regulation of their HSE containing reporter.Taken by itself, ReportSeq represents a clear and important step forward in genetic screening methodologies, combining the ease and genetic specificity of CRISPRi screens with a more nuanced phenotypic readout than growth. The limitations of the method should be more clearly spelled out. Some technical details of the implementation could use more exploration, such as the effect of multi-day activation of gene repression and the effectiveness of various sgRNAs, but the authors clearly demonstrate the usefulness of the method. The novel insights into the HSR, however, are less convincing. Similar screens have been carried out by Onn Brandman using a fluorescence based reporter and synthetic gene arrays (Brandman et al., Cell 2012) and the findings are largely in agreement. The broader significance of particular cases highlighted in the paper would benefit greatly from validation beyond the synthetic reporter used in this study. In addition, the interaction scores used to quantify the effects of gene knockdown are influenced by the extent to which each gene regulates the basal HSR, which complicates the interpretation of some of the results.1. In the introduction, "Additionally, post-transcriptional models have been proposed, such as HSR-regulated mRNA half life (Heikkinen, 2003) or regulation of the translation efficiency of heat shock mRNAs (Zid and O'Shea, 2014). Yet the extent to which these and other mechanisms drive the HSR under diverse stressors is poorly understood…" This section introduces the HSR as a transcriptional response, and states that it is driven by Hsf1, but mRNA half life or translational efficiency are not thought to regulate or activate Hsf1 or the transcriptional response; these are downstream of the HSR. Later material in the manuscript indicates that "regulation of the HSR" includes translation, not merely transcription as the introduction states. The writing should be clarified.

We have clarified the introduction to state that the HSR drives gene expression (rather than just being a transcriptional response). Thank you for bringing this issue to our attention.

2. Please provide more detail on the reporter construct used. It is suggested that it is a combination of HSEs and the CYC1 promoter as used in Brandman et al., 2012. However the authors say only that it is "similar" to that reporter without specifying the ways in which it is different. Some additional discussion of the extent to which this reporter is regulated by Hsf1 vs other stress-induced transcription factors such as Msn2/4 is important for interpreting the results of the paper. For instance, any Hsp70-independent regulation could either be a result of Hsp70-independent regulation of Hsf1 or non-Hsf1 regulation of the reporter. I recommend additional experiments be done combining CRISPRi knockdown of key genes such as ASC1 and HSC82 with additional reporter constructs, either with promoters known to be regulated by Hsf1 or Msn2/4 or with other synthetic promoters containing STRE elements.

The reporter used in this study is in fact identical to that used itn Brandman et al., 2012. The only difference that the study employs the reporter on a 2u plasmid while Brandman et al., 2012. integrated the reporter. This has been clarified in the text. There are no msn2 binding sites on this minimal promoter, but it is possible that regulators of the cryppled cyc1 sequence will appear as hits in the screen. This is now stated in the text. To verify that top hits in the screen are regulating the Hsf1 rather than other elements in the synthetic promoter, we tested deletion strains of 3 top hits using a reporter identical to our HSR reporter except without the Hsf1 binding site. None of the hits activated the control reporter (Figure S2-2). Additionally, another publication from our lab shows that deleting two top hits from our screen (SSA2 and HSP104) does not activate a different reporter identical to our Hsf1 reporter except with Hsf1 binding sites replaced by those for Rpn4 (Work and Brandman JCB 2020, Figure 4F).

3. The analysis would benefit from a multiple-testing correction. What the authors call "confidence" is effectively statistical significance of (replicated) scores under a normal null model, a Z score, which has an associated probability. We urge the authors to apply a false discovery rate (FDR) correction, or their choice of multiple-testing correction, to assess the adjusted probability that the observed scores occur by chance in a dataset of this size.

Thanks for bringing this up--this is a nuanced issue. First of all, we should not have been referring to these as ''confidence scores'', since they do not actually correspond to the confidence of rejecting the null hypothesis [no modulatory role] in any rigorous sense. In our revision, we will simply refer to them as scores. The results would not change significantly if we were to apply an FDR correction (since the corrected scores would preserve the order of the genes). Still, we don't believe that such a correction would make sense here. FDR corrections are most applicable in settings in which one wishes to understand the confidence with which one can reject the hypothesis that all trials correspond to the null hypothesis. In our settings, we do know that some genes have modulatory roles (e.g. the regulatory genes, and genes known to be associated with these). We would not want to apply an FDR correction to the scores of the control genes, because the identity of these genes was fixed prior to the experiment (there is no 'multiple hypothesis' issue for these genes). Additionally, some of the ways in which we use the scores (e.g. in building gene ontologies, looking at correlations between replicates and conditions, etc.) is different from the settings in which these corrections make the most statistical sense (i.e. when arguing that a single hypothesis is significant).

4. The induction of the reporter after knockout of gene X is interpreted as meaning that X regulates the HSR. However, knocking out X could (say) cause buildup of a toxic intermediate in some pathway, triggering the cellular stress response and thus activation of the HSR. X here would not be a "regulator" of the transcriptional response in the sense in which the authors intend. The logic of the connection between signal and interpretation should be more clearly spelled out and its limitations or edge cases mentioned.

This limitation is now explicitly stated in the discussion.

5. More technical details in the Methods section would be valuable to the community. For instance a diagram of the cloning process would be useful. In addition, providing the sequences of the adapters and primers and the times and concentrations of enzymes used in library preparation would allow for replication of the technique.

We have included a more detailed explanation of the plasmid library design strategy, including sequences, used in this study (see description in Methods and Supplemental data file 8 contents library_creation_diagram, library_primer_sequences, and library_sequences).

6. The inclusion of the code used and the pseudocode description of it is greatly appreciated. One question that remained is how the biological replicates are used in the creation of the score. The only time that the replicates are used or shown are in the volcano plots in Figure 2C. Supplemental figures showing the reproducibility across replicates or a description of how the replicates were used in the scoring process would be helpful.

Replicates were averaged using arithmetic mean of the normalized scores. In addition to the volcano plot of basal scores, replicates for stress conditions and including R values are now shown in figure S5-1. Note that stress conditions with few interactors have low R values and those with many interactors high higher R values.

7. In Figure S1A no induction of the reporter is seen with a SSA2 targeting sgRNA. This is at odds with subsequent data showing a strong effect of SSA2 knockdown. Presumably this is a result of poor efficacy of the chosen sgRNA and additional sgRNAs should be tested to resolve this discrepancy. Furthermore, it would be useful to know the extent of knockdown, for instance by tracking SSA2 mRNA levels by qPCR.

In Figure S1A, 0 means no induction and 1 means double basal levels. So the SSA2-targeting sgRNA robustly induces the HSR reporter (almost two-fold) as expected.

8. Later, the authors point out the importance of having multiple guide RNAs and barcodes for each gene. It would be helpful to see how important these factors are by splitting out the results for a select set of genes (perhaps CYC1, SIS1 and SSA2) into guide RNA- and barcode- specific effects.

As requested, we have now broken down sgRNA and barcode scores for CYC1, SIS1 and SSA2 (Figure S2-1).

9. One drawback to the approach used here is that the library is grown for multiple days with active CRISPRi targeting of genes. Presumably this could lead to the depletion of essential genes from the pool and a discussion of the extent to which this depletion was observed would provide important context for the breadth of the screen.

We now show that essential genes were weakly depleted (Figure S1-1D). However, these still are a rich source of as hits (e.g. SIS1)

10. The authors mention that sgRNAs targeting the reporter promoter (CYC1 sgRNAs) do not prevent its activation during heat stress. This implies that knockdown under basal conditions does not necessarily prevent the induction of genes under heat shock conditions. This has important ramifications for the interpretation of later time points of the heat shock time course, partly in regards to heat shock induced genes, as the differential effects seen could result from changes in the extent of the knockdown. Western blots showing the extent of depletion for a subset of genes (such as SSA2, SSA1, SIS1 etc.) with selected guide RNAs during the time course would alleviate this concern.

We tested if heat shock affects CRISPR knockdown efficiency using qPCR for SSA2 and SIS1 mRNA during a heat shock time course (Figure S4-1). No change is either mRNA knockdown was observed throughout the time course, demonstrating that heat shock does not affect CRISPR knockdown efficiency.

11. The identification of differences in the roles of SSA1 and SSA2, and Sis1 as opposed to Ydj1 are both important contributions. However, in order to demonstrate aprove a difference, it is necessary to show that these genes are all being knocked down to a similar extent. For example, SSA2 is induced by heat shock far less than SSA1, and the authors observe a time-dependent effect of SSA1 but not SSA2, raising the question as to what extent SSA1's induction profile has been altered by knockdown. Thus weI suggest additional experiments showing differential effects on HSR induction using specific guide RNAs with western blots or qPCR to show the extent of depletion. Alternatively the authors could note that they have no evidence for the knockdown of these genes during the timecourse or for the relative levels of the knockdowns.

To confirm the difference in SSA1 vs SSA2 phenotypes, we now measure the effects of knockout of these genes on an integrated HSR reporter (Figure S5-2). We observed basal HSR levels and synergy with AZC addition consistent with the screen results. We also now mention in the text that we do not measure knockdown efficiency of each gene and cannot rule out the effect of any gene on the HSR based solely on the screen. Note that a strong interaction of at least one scressor for a gene suggests that the knockdown worked to some degree, so that for that gene low scores in other conditions may be informative.

12. Because the interaction score used is based on the basal levels of regulation, genes which strongly affect the basal response such as Sis1 are expected to have strong negative interaction scores. Thus it seems that both the plots shown in Figure 3C and the PCA analysis in Figure S2 could be largely explained by the basal score of genes such as SIS1, SSA2 and SSE1. Some more discussion of how to disentangle genetic interactions with stress from the effects of knockdown on the basal HSR would be very helpful for interpreting these results.

The reviewer raises an interesting point. However, SIS1 had a low basal score yet had a high PCA score. Therefore basal scores do not completely limit interaction scores and the PCA can pick up on meaningful interactions.

13. Figure 6 does not convincingly show that GCN3 knockdown selectively affects translation of the reporter as opposed to a more general effect. Additional experiments as in Figure 6B but with reporters containing different promoters and 5' UTRs would help prove that the reporter is unique in its response to GCN3 knockdown.

Thank you for bringing this up. To address this concern, we attempted to conduct experiments using a GAL1 promoter to regulate the expression of the GFP reporter and performing CRISPRi knockdown of GCN3. However, we were unable to obtain consistent results in these conditions, since cells exhibit severe slow growth in different carbon sources (raffinose and galactose) on top of expressing two different episomal plasmids (reporter and dCas9) (data not shown). We then proceeded to construct two additional reporters, one consisting of the full-length promoter of HSP12 (-965 bp) controlling GFP, and a truncated promoter of HSP12 (-805 bp). The truncation of the HSP12 promoter removed the only HSE found in its sequence, while preserving the 7 STRE. It is worth mentioning that both reporters are identical in its 5’ UTR, CDS and 3’ UTR. Removal of the single HSE of prHSP12 does not seem to be essential for its expression in heat stress conditions (Chowdhary et al., 2019), and after sodium arsenite and AZC treatments (new Figure 6A). These findings suggest that Hsf1 is dispensable for HSP12 expression in a variety of stress conditions and that these reporters can be used to answer whether GCN3 knockdown is selective to our Hsf1-responsive synthetic promoter. We observed that GCN3 knockdown affects the translation efficiency of all the reporters during arsenite stress, but not in AZC (new Figure 6). Our experiments suggest that GCN3 increases translation efficiency of stress induced genes including those with and without an HSE element. Future studies will determine how translation of non-stress induced genes is affected by GCN3 in arsenite stress.